# SplitGaussian: Reconstructing Dynamic Scenes via Visual Geometry Decomposition

## Abstract

Reconstructing dynamic 3D scenes from monocular video remains fundamentally challenging due to the need to jointly infer motion, structure, and appearance from limited observations. Existing dynamic scene reconstruction methods based on Gaussian Splatting often entangle static and dynamic elements in a shared representation, leading to motion leakage, geometric distortions, and temporal flickering. We identify that the root cause lies in the coupled modeling of geometry and appearance across time, which hampers both stability and interpretability. To address this, we propose **SplitGaussian**, a novel framework that explicitly decomposes scene representations into static and dynamic components. By decoupling motion modeling from background geometry and allowing only the dynamic branch to deform over time, our method prevents motion artifacts in static regions while supporting view- and time-dependent appearance refinement. This disentangled design not only enhances temporal consistency and reconstruction fidelity but also accelerates convergence. Extensive experiments demonstrate that SplitGaussian outperforms prior state-of-the-art methods in rendering quality, geometric stability, and motion separation.

## 1 Introduction

Reconstructing dynamic 3D scenes from monocular video remains a core challenge in computer vision, with far-reaching applications in virtual reality, free-viewpoint rendering, and autonomous perception. These scenes present non-rigid motion, occlusions, and appearance variation, demanding joint inference of structure, motion, and camera pose from limited visual cues. Traditional multi-view stereo and depth sensors offer stronger constraints but restrict flexibility. Methods like NR-NeRF (Tretschk et al., 2021) introduce a canonical volume plus deformation field to enable dynamic reconstruction from monocular video, but they require expensive per-scene ray-based optimization and converge slowly.

Compared to implicit volumetric fields, 3D Gaussian Splatting (3DGS) (Kerbl et al., 2023) offers explicit and compact representations, enabling faster optimization and real-time rendering. Early attempts at dynamic reconstruction with 3DGS treat each frame independently by reconstructing a separate set of Gaussians per frame (Luiten et al., 2024b), which fails to establish temporal consistency and leads to redundant or unstable representations. To this end, existing cutting-edge dynamic extensions—such as Deformable 3DGS (Yang et al., 2023b)—use a unified deformation network across static and dynamic regions, often disturbing static structures and introducing temporal artifacts. When a single deformation field is applied uniformly across both dynamic and static regions, it tends to propagate motion artifacts into static areas, leading to geometric distortions (e.g., *rigid structures slightly shifted or warped*) and appearance inconsistencies (e.g., *temporal texture flicker or color drift*). This issue also persists in prior works (Wu et al., 2024; Lu et al., 2025; Kwak et al., 2025; Katsumata et al., 2024), where static scene elements exhibit lingering artifacts despite motion modeling.

Let us revisit the Gaussian Splatting process, where each primitive jointly encodes **visual appearance** (e.g., color defined by spherical harmonic coefficients, opacity) and **geometry** (e.g., position, rotation, and scaling). We argue that such joint modeling underlies many common artifacts in dynamic reconstruction, such as motion leakage into static regions and temporal inconsistencies. Existing methods like DeGauss (Wang et al., 2025) (depth-aware compositing), DynaSplat (Deng et al.,

2025) (optical flow guidance), and GauFre (Liang et al., 2025) (multi-stream processing) attempt to address dynamics, but all suffer from effectively disentangling dynamic and static regions from the underlying meta-representations.

Motivated by this, we propose to decompose the visual geometry within the gaussian representation to explicitly model dynamic and static regions. Specifically, the static part maintains fixed geometry but allows appearance to vary over time, while the dynamic part models time-varying geometry and appearance through a deformation network conditioned on shared spatiotemporal encodings. This design effectively decouples motion modeling from background representation, yielding more robust and interpretable reconstructions. We further improve reconstruction by applying visibility-driven pruning to remove low-contribution static Gaussians and introducing a depth-aware pretraining phase for better geometric initialization and depth consistency. Our approach yields more stable optimization, temporally consistent reconstructions, and superior visual quality compared to existing methods that use a single deformation field for the entire scene. We summarize the key contributions as follows:

- We introduce an explicit decomposition of Gaussian primitives into static and dynamic components, enabling disentangled modeling of geometry and appearance to improve reconstruction stability.
- We orchestrate a unified framework with shared spatiotemporal encoding, dedicated deformation network, and visibility-driven pruning for efficient and coherent dynamic scene reconstruction from monocular video.
- Extensive experiments demonstrate that our method achieves superior performance in reducing geometric distortions and appearance flickering, outperforming existing state-of-the-art baselines.

## 2 RELATED WORK

### 2.1 DYNAMIC SCENE RECONSTRUCTION

Dynamic scene reconstruction seeks to recover geometry and appearance under challenging conditions such as occlusions and illumination changes. Traditional methods, including multi-view stereo (Newcombe et al., 2015) and scene flow (Vogel et al., 2013), require dense depth and struggle with significant deformations or fast motion. Learning-based approaches (Ma et al., 2019) jointly predict geometry and motion from monocular inputs but often lack temporal coherence. Recent neural implicit methods, such as NeRF (Mildenhall et al., 2020), utilize continuous volumetric representations with deformation fields (D-NeRF (Pumarola et al., 2020), NSFF (Li et al., 2021)) or higher-dimensional embeddings (HyperNeRF (Park et al., 2021)) for dynamic reconstruction. Despite their high visual fidelity, these methods involve costly per-scene optimization and slow inference, limiting real-time application. In contrast, 3DGS (Kerbl et al., 2023) employs rasterization-based anisotropic Gaussians, enabling efficient optimization and real-time rendering. Recent dynamic extensions (Yang et al., 2023b; Wu et al., 2024; Lu et al., 2025; Kwak et al., 2025; Deng et al., 2025) incorporate time-varying transformations into Gaussian to improve spatiotemporal consistency.

### 2.2 DECOMPOSITION OF DYNAMIC AND STATIC REGION

Decomposing scenes into static and dynamic components simplifies motion modeling and enhances reconstruction quality. Early methods such as DeGauss (Wang et al., 2025) and DynaSplat (Deng et al., 2025) employ external motion cues or optical flow-based masks, often using separate branches or losses. GauFre (Liang et al., 2025) further introduces dual-branch architectures with occlusion reasoning to improve temporal coherence. CoGS (Yu et al., 2024) uses compositional modeling with learned masks for flexible blending, but at the cost of increased complexity. BARD-GS (Lu et al., 2025) introduces deformation modeling into static regions, addressing motion blur but risking static geometry distortion. These methods commonly face limitations such as dependency on external priors, architectural complexity, or unintended static-region deformation. In contrast, our method explicitly decomposes geometry and appearance within Gaussian primitives through unified spatiotemporal encoding. We avoid separate encoders and complex fusion modules, maintain fixed

Figure 1: **Framework Overview**. We adopt a two-stage training pipeline: **Stage I** disentangles static and dynamic Gaussians via region-specific supervision and visibility-driven pruning to enhance geometric stability; **Stage II** jointly optimizes both components, where static appearance is modeled without deformation and dynamic motion is learned via a spatiotemporally-conditioned deformation network, enabling mutual refinement and improved reconstruction fidelity.

geometry for static Gaussians with residual temporal appearance modeling, and employ visibility-based pruning and depth-aware pretraining, significantly enhancing stability, temporal consistency, and realism.

## 3 METHOD

We propose a dynamic scene reconstruction framework that explicitly decomposes geometry into static and dynamic components, modeled via a unified spatiotemporal encoding. Static Gaussians maintain fixed positions with time-varying appearance, while dynamic Gaussians undergo learned motion-based deformation. Reconstruction is supervised through region-specific losses guided by visibility masks, ensuring temporal consistency and disentanglement. A visibility-driven pruning strategy improves static reliability and efficiency, and depth-aware pretraining further refines geometry alignment. We will detail this later.

### 3.1 PRELIMINARY: 3D GAUSSIAN SPLATTING

3D Gaussian Splatting (Kerbl et al., 2023) represents a scene using a set of anisotropic Gaussians, each defined by a 3D center $\mu$, covariance matrix $\Sigma$, spherical harmonic (SH) color coefficients $C$, and opacity $\alpha$. The Gaussian density at a point $X$ is given by:

$$G(X) = \exp\left(-\frac{1}{2}X^\top \Sigma^{-1} X\right). \tag{1}$$

For efficient optimization and interpretation, the covariance matrix is typically decomposed as:

$$\Sigma = RSS^\top R^\top, \tag{2}$$

where $R$ is a rotation matrix and $S$ is a scaling matrix.

During rendering, the Gaussian is projected into screen space by applying the viewing transformation matrix $W$ and the Jacobian matrix $J$ of the affine approximation of the camera projection:

$$\Sigma' = JW\Sigma W^\top J^\top. \tag{3}$$

The final pixel color is computed via alpha compositing in front-to-back order as:

$$C = \sum_{i=1}^{N} c_i \alpha_i \prod_{j=1}^{i-1}(1 - \alpha_j), \tag{4}$$

where $c_i$ and $\alpha_i$ denote the color and opacity of the $i$-th Gaussian. This representation enables real-time rendering with high visual fidelity, but it inherently assumes static scene geometry. When extended to dynamic scenes, naively optimizing a shared set of Gaussians often leads to motion artifacts and temporal inconsistencies, due to the entanglement of geometry and appearance modeling. These challenges motivate our decomposition-based formulation, which explicitly separates static and dynamic components to ensure more stable and interpretable reconstructions.

## 3.2 Visual Geometry Decomposition

A core challenge in dynamic scene reconstruction is simultaneously modeling time-varying geometry and appearance. To address this, we explicitly decompose the scene at time $t$ into two sets of Gaussian primitives:

$$G(t) := \{G_{\text{s}}(\mu_s, \Sigma_s, w_s(t))\} \cup \{G_{\text{d}}(\mu_d(t), \Sigma_d(t), w_d(t))\}, \tag{5}$$

where each Gaussian primitive $G(\mu, \Sigma, w)$ consists of:

- **Appearance**: represented by attributes $w$, including spherical harmonic coefficients and opacity.
- **Geometry**: represented by center position $\mu$ and covariance matrix $\Sigma$.

In our formulation:

- The static component $G_{\text{s}}$ maintains fixed geometry $(\mu_s, \Sigma_s)$ and only allows temporal variation in appearance $w_s(t)$.
- The dynamic component $G_{\text{d}}$ models both geometry deformation $(\mu_d(t), \Sigma_d(t))$ and appearance variation $(w_d(t))$ over time.

This explicit separation of geometry and appearance preserves static background integrity and accurately isolates dynamic regions.

**Unified Spatiotemporal Encoding.** We adopt a unified sinusoidal positional encoding $\gamma(\cdot)$ to ensure consistent parameterization across both the geometry deformation and appearance modeling modules. Specifically, given the 3D Gaussian center position $\mu \in \mathbb{R}^3$ and a scalar time $t$, the encoding is defined as:

$$\gamma(p) = \left(\sin(2^k \pi p), \cos(2^k \pi p)\right)_{k=0}^{L-1}, \tag{6}$$

where $p$ represents either a spatial coordinate $(\mu_x, \mu_y, \mu_z)$ or the temporal scalar $t$, and $L$ controls the number of frequency bands. The combined input feature for subsequent modules is thus constructed as:

$$[\gamma(\mu), \gamma(t)]. \tag{7}$$

This encoding is consistently shared across both the deformation MLP and the residual appearance MLPs. Empirically, we set $L = 10$ for spatial coordinates and $L = 6$ for temporal encoding in synthetic scenes, while using $L = 10$ for both dimensions in real-world scenarios.

**Static Component.** We represent static regions using Gaussian primitives $\mathcal{N}_s(\mu_s, \Sigma_s)$, whose geometry $(\mu_s, \Sigma_s)$ is fixed over time. To model temporal variations in appearance, such as illumination changes, we predict a residual to the initial (frozen) appearance parameters:

$$w_{s,i}(t) = w_{s,i}^{(0)} + \Delta w_{s,i}(t) \tag{8a}$$

$$\Delta w_{s,i}(t) = \text{MLP}_{\text{app}}^{(s)}\left([\gamma(\mu_{s,i}), \gamma(t)]\right), \tag{8b}$$

where $w_{s,i}^{(0)}$ denotes the initial spherical harmonic (SH) coefficients and opacity, which remain fixed, and $\Delta w_{s,i}(t)$ is the residual predicted by the appearance MLP. Thus, each static Gaussian can temporally adapt its appearance without altering its geometry. We supervise the static reconstruction using a combination of L1 and Structural Similarity (SSIM) losses, computed within a binary static-region mask $\mathbf{M}$:

$$\mathcal{L}_{\text{static}} = \mathcal{L}_1(\hat{I}_s \odot \mathbf{M}, I_{\text{gt}} \odot \mathbf{M}) + \lambda_{\text{ssim}} \cdot (1 - \text{SSIM}(\hat{I}_s \odot \mathbf{M}, I_{\text{gt}} \odot \mathbf{M})), \tag{9}$$

where $\hat{I}_s$ is the rendered static image, and $I_{\text{gt}}$ is the ground-truth reference.

**Dynamic Component.** A straightforward baseline for modeling dynamic content is to optimize separate Gaussian primitives per timestamp and interpolate post-hoc (Luiten et al., 2024a). However, such decoupled modeling lacks temporal coherence and cannot effectively represent continuous motion patterns. Real-world motions, in contrast, are typically temporally continuous and locally smooth, suggesting that a deformation-based formulation is more suitable. Motivated by recent advances (Yang et al., 2023b), we employ a deformation-based formulation. Specifically, we introduce a deformation network, parameterized by $\theta$, which predicts time-dependent offsets to canonical Gaussian parameters. Given a canonical position $\mu_d(0)$ and a timestamp $t$, the deformation network outputs offsets for position, scale, and rotation:

$$(\delta\mu, \delta r, \delta s) = \mathcal{F}_\theta\big([\gamma(\mu_d(0)), \gamma(t)]\big). \tag{10}$$

These offsets update the Gaussian's geometry at time $t$ as:

$$\mu_d(t) = \mu_d(0) + \delta\mu, \tag{11a}$$

$$\Sigma_d(t) = \mathbf{A}_d(t)\,\Sigma_d(0)\,\mathbf{A}_d^\top(t) \tag{11b}$$

where $\mathbf{A}_d(t)$ is derived from $\delta r, \delta s$ and governs the anisotropic scaling and orientation. This formulation enables temporally smooth and flexible deformation modeling through shared spatiotemporal encoding.

The dynamic component is supervised via region-specific losses computed over the dynamic mask region $\mathbf{1} - \mathbf{M}$:

$$\mathcal{L}_{\text{dynamic}} = \mathcal{L}_1(\hat{I}_d, I_{\text{gt}} \odot (\mathbf{1} - \mathbf{M})) + \lambda_{\text{ssim}} \cdot (1 - \text{SSIM}(\hat{I}_d, I_{\text{gt}} \odot (\mathbf{1} - \mathbf{M}))) \tag{12}$$

where $\hat{I}_d$ is the dynamically rendered image. This mask-based supervision ensures each Gaussian module receives gradients exclusively from its corresponding visible regions, promoting stable training and effective disentanglement between static and dynamic components.

**Remark I. Beyond Prior Decomposition Schemes.** Recent works (Lu et al., 2025; Liang et al., 2025; Yu et al., 2024; Wang et al., 2025; Deng et al., 2025)share the similar sprit of decomposing scenes into dynamic and static parts via auxiliary cues like masks, optical flow, or multi-branch networks. While effective, these methods often suffer from entangled representations, causing motion leakage and unstable training. In contrast, our method performs explicit geometry-level decomposition within the 3D Gaussian representation. Static Gaussians maintain fixed geometry with time-varying appearance, while dynamic Gaussians deform via a shared spatiotemporal encoding. This unified design eliminates the need for dual-stream architectures or occlusion modeling, improving both efficiency and stability. Combined with visibility-driven pruning and depth-aware initialization, our framework achieves disentangled, temporally coherent reconstructions across diverse scenes.

**Remark II. Mask-Guided Disentangled Optimization.** To segment dynamic regions, we employ explicit per-frame binary masks $\mathbf{M} \in \{0, 1\}$ generated by the open-vocabulary tracker *Track Anything* (Yang et al., 2023a). These masks supervise static and dynamic Gaussians separately—using $\mathbf{M}$ to ensure robust structural disentanglement. Unlike prior methods (Wang et al., 2025; Deng et al., 2025; Liang et al., 2025) that rely on optical flow or learned soft masks for implicit separation, our approach, akin to BARD-GS (Lu et al., 2025), benefits from externally provided masks for more accurate and temporally consistent guidance. To further enhance disentanglement, we introduce an asymmetric masking strategy: for static regions (Eq. (9)), both prediction and ground truth are masked by $\mathbf{M}$, while for dynamic regions (Eq. (12)), only the ground truth is masked. This preserves occluded static geometry while enabling complete reconstruction of dynamic content for more stable training and improved reconstruction quality.

### 3.3 VISIBILITY-DRIVEN PRUNING

In dynamic scene reconstruction, static Gaussians located near view boundaries or occluded regions are often weakly supervised due to limited visibility, leading to unstable optimization and redundancy. To address this, we propose a visibility-driven pruning strategy that quantifies each static Gaussian's long-term contribution over the video sequence. Unlike SplaTAM (Keetha et al., 2024), which employs per-frame RGB-D silhouettes for pruning and densification, our method accumulates

visibility and opacity across time, ensuring that only Gaussians with persistently low contribution are removed, thereby stabilizing static geometry under monocular supervision.

Let $G_{\mathrm{s}} = \{G_{s,i}(\mu_{s,i}, \Sigma_{s,i}, w_{s,i}(t))\}_{i=1}^{N_s}$ denote the set of static Gaussians. For each $G_{s,i} \in G_{\mathrm{s}}$, we define its integrated visibility score as:

$$\bar{V}_{s,i} = \frac{1}{T} \sum_{t=1}^{T} \mathbb{1}_{\{G_{s,i} \text{ rendered at } t\}} \cdot (1 - \alpha_{s,i}(t)) \tag{15}$$

where $T$ is the total number of training frames, $\mathcal{V}_t^{(s)} \subseteq G_{\mathrm{s}}$ denotes the subset of visible static Gaussians at time $t$, $\mathbb{1}_{\{G_{s,i} \text{ rendered at } t\}}$ is the indicator function (1 if $G_{s,i}$ is rendered at time $t$), $\alpha_{s,i}(t)$ is the opacity of $G_{s,i}$ at time $t$.

This formulation unifies notation with earlier definitions and reflects both temporal visibility and opacity modulation. We prune low-contribution Gaussians based on $\bar{V}_{s,i}$ thresholds to improve training stability and reduce redundancy.

### 3.4 DEPTH-AWARE PRETRAINING

3D Gaussian Splatting (3DGS) (Kerbl et al., 2023) demonstrates that initializing Gaussian primitives using Structure-from-Motion (SfM) (Schonberger & Frahm, 2016) point clouds can facilitate effective training. However, we find that such initializations are often suboptimal for depth-guided reconstruction, frequently leading to geometric inconsistencies. This issue stems from the requirement to align the SfM reconstruction with available depth images by estimating a global scale and offset. In practice, inaccuracies or sparsity in the SfM point cloud can impair this calibration, thereby weakening the efficacy of depth-based regularization. To address this, we introduce a short pretraining stage before the main optimization. This stage refines the initial geometry and improves alignment between the scene structure and depth supervision, resulting in a more reliable point cloud for subsequent learning. Specifically, we regularize the static Gaussian geometry using monocular depth maps through the following loss:

$$\mathcal{L}_{\mathrm{depth}} = \lambda_{\mathrm{depth}}(t) \cdot \left\| (\hat{D} - D_{\mathrm{gt}}) \odot \mathbf{M} \right\|_1 \tag{14}$$

where $\hat{D}$ denotes the rendered depth map from the static Gaussians, $D_{\mathrm{gt}}$ is the ground-truth monocular depth map, and $\mathbf{M}$ is the binary mask identifying static regions. The time-dependent decay factor $\lambda_{\mathrm{depth}}(t)$ gradually reduces the influence of depth supervision as training progresses. This pretraining stage enhances geometric consistency and stabilizes later optimization.

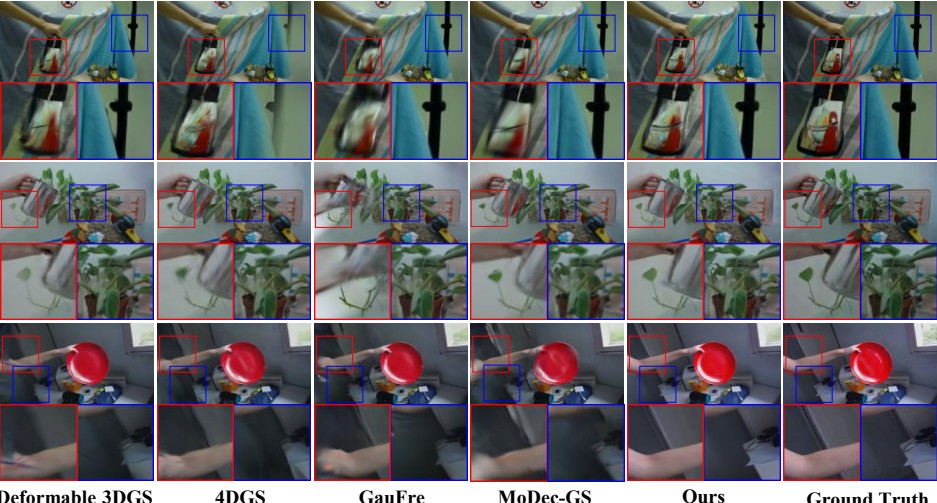

|   Deformable 3DGS   |   4DGS   |   GauFre   |   MoDec-GS   |   Ours   |   Ground Truth   |

Figure 2: Qualitative comparisons on the NeRF-DS (Yan et al., 2023) monocular video dataset. Red and blue boxes highlight regions where our method notably improves visual quality compared to prior approaches.

## 3.5 TRAINING PROTOCOL.

Our training pipeline follows a two-stage design with an initial *Depth-Aware Pretraining (DAP)* phase to refine SfM-initialized geometry.

**Stage I (S1)** focuses on disentangling static and dynamic components via region-specific supervision using binary masks $\mathbf{M}$. Static Gaussians are optimized with pixels where $\mathbf{M} = 1$, while dynamic Gaussians use the complementary region. To further improve static geometry quality, we introduce a *Visibility-Driven Pruning (VDP)* strategy that removes low-visibility static Gaussians (e.g., near view frustums or edges), thereby mitigating supervision noise and improving geometric stability.

**Stage II (S2)** performs joint optimization of both components. Static Gaussians retain fixed geometry and are refined using an **Appearance Model (APP)**, which predicts view- and time-dependent appearance (e.g., color and opacity) without invoking deformation. Dynamic Gaussians are updated through a learned deformation network conditioned on shared spatiotemporal encodings. During this stage, gradients propagate across both static and dynamic components, allowing mutual refinement. This two-stage design significantly improves reconstruction fidelity and training stability, as validated in our ablation studies (Table 3), more details can be seen in Figure 1.

# 4 EXPERIMENT

## 4.1 EXPERIMENTAL SETUP

We implement our method in PyTorch (Paszke et al., 2019), building upon the official 3D Gaussian Splatting (Kerbl et al., 2023) and Deformable 3DGS (Yang et al., 2023b) frameworks. Training is conducted in two stages: first, we separately optimize static and dynamic components for 30k iterations without the static appearance model; second, we jointly train both components with appearance modeling for 40k iterations. A single Adam optimizer (Adam et al., 2014) with $\beta_1 = 0.9$, $\beta_2 = 0.999$, and a learning rate decaying exponentially from $8 \times 10^{-4}$ to $1.6 \times 10^{-6}$ is used. Both deformation and appearance MLPs share architecture and schedule. Visibility-driven pruning removes rarely rendered Gaussians to improve training efficiency. For depth-supervised scenes, a short pretraining stage aligns SfM point clouds with depth maps before applying depth regularization. All experiments are conducted on a single NVIDIA 5090 GPU (32GB). Training a typical sequence takes around 1 hour, with peak memory usage below 30 GB, and inference runs at 200 FPS for a resolution of 480×270.

We evaluate our method on three datasets: (1) **iPhone** (Gao et al., 2022), featuring 14 real-world scenes (4180 frames at $720 \times 480$) with handheld motion and diverse dynamics; (2) **NeRF-DS** (Yan et al., 2023), a monocular dataset containing specular objects and challenging motion; and (3) **HyperNeRF** Park et al. (2021), which includes complex dynamic scenes captured in real-world environments. Following standard protocols, we report PSNR, SSIM, and LPIPS (Zhang et al., 2018) to evaluate novel view synthesis quality.

| | *as* | | | *basin* | | | *bell* | | | *cup* | | |
|---|---|---|---|---|---|---|---|---|---|---|---|---|
| Method | PSNR ↑ | SSIM ↑ | LPIPS ↓ | PSNR ↑ | SSIM ↑ | LPIPS ↓ | PSNR ↑ | SSIM ↑ | LPIPS ↓ | PSNR ↑ | SSIM ↑ | LPIPS ↓ |
| HyperNeRF$_{\text{SIGGRAPH Asia 2021}}$ | 25.59 | 0.8567 | 0.1754 | 20.41 | 0.8099 | 0.1889 | 23.06 | 0.7698 | 0.2479 | 23.98 | 0.8531 | 0.1988 |
| NeRF-DS$_{\text{CVPR 2023}}$ | 25.34 | 0.8679 | 0.1515 | 20.23 | 0.8032 | 0.2008 | 22.57 | 0.7821 | 0.2489 | 24.51 | 0.8659 | 0.1668 |
| Deformable 3DGS$_{\text{CVPR 2024}}$ | 26.03 | 0.8836 | 0.1351 | 19.67 | 0.7867 | 0.1498 | 24.48 | 0.7997 | 0.1822 | 24.50 | 0.8763 | 0.1472 |
| 4DGS$_{\text{CVPR 2024}}$ | 24.77 | 0.8642 | 0.1521 | 19.36 | 0.7677 | 0.1678 | 23.16 | 0.8015 | 0.1571 | 23.88 | 0.8691 | 0.1532 |
| GauFre$_{\text{WACV 2025}}$ | 26.05 | 0.8790 | 0.1244 | 19.54 | 0.7780 | 0.1222 | 25.24 | 0.8130 | 0.1351 | 24.04 | 0.8191 | 0.2054 |
| MoDec-GS$_{\text{CVPR 2025}}$ | 24.65 | 0.8538 | 0.1460 | 19.57 | 0.7787 | 0.1805 | 22.19 | 0.7562 | 0.2312 | 24.18 | 0.8798 | 0.2643 |
| **Ours** | 26.01 | 0.8806 | 0.1031 | 19.78 | 0.7885 | 0.1278 | 25.55 | 0.8484 | 0.1425 | 24.63 | 0.8829 | 0.1112 |

| | *plate* | | | *press* | | | *sieve* | | | **Average** | | |
|---|---|---|---|---|---|---|---|---|---|---|---|---|
| Method | PSNR ↑ | SSIM ↑ | LPIPS ↓ | PSNR ↑ | SSIM ↑ | LPIPS ↓ | PSNR ↑ | SSIM ↑ | LPIPS ↓ | **PSNR ↑** | **SSIM ↑** | **LPIPS ↓** |
| HyperNeRF$_{\text{SIGGRAPH Asia 2021}}$ | 21.10 | 0.7979 | 0.2614 | 24.59 | 0.8263 | 0.2385 | 25.41 | 0.8593 | 0.2142 | 23.44 | 0.8247 | 0.2178 |
| NeRF-DS$_{\text{CVPR 2023}}$ | 19.70 | 0.7813 | 0.2467 | 25.34 | 0.8711 | 0.2032 | 24.99 | 0.8705 | 0.2067 | 23.24 | 0.8345 | 0.2035 |
| Deformable 3DGS$_{\text{CVPR 2024}}$ | 19.88 | 0.8293 | 0.1914 | 25.32 | 0.8752 | 0.1378 | 25.62 | 0.8627 | 0.1206 | 23.64 | 0.8447 | 0.1520 |
| 4DGS$_{\text{CVPR 2024}}$ | 18.77 | 0.7891 | 0.1857 | 24.81 | 0.8311 | 0.1598 | 25.16 | 0.8611 | 0.1234 | 22.84 | 0.8262 | 0.1570 |
| GauFre$_{\text{WACV 2025}}$ | 20.00 | 0.8051 | 0.2323 | 25.05 | 0.8545 | 0.1763 | 24.88 | 0.8568 | 0.1623 | 23.54 | 0.8293 | 0.1654 |
| MoDec-GS$_{\text{CVPR 2025}}$ | 18.87 | 0.7306 | 0.2547 | 22.87 | 0.7296 | 0.2111 | 23.48 | 0.7982 | 0.2001 | 22.25 | 0.7895 | 0.2125 |
| **Ours** | 20.34 | 0.8116 | 0.1413 | 25.43 | 0.8701 | 0.1498 | 26.49 | 0.8753 | 0.1137 | 24.03 | 0.8510 | 0.1270 |

Table 1: Quantitative results comparison on the NeRF-DS (Yan et al., 2023) dataset. Red and orange cells denote the best and second-best results, respectively.

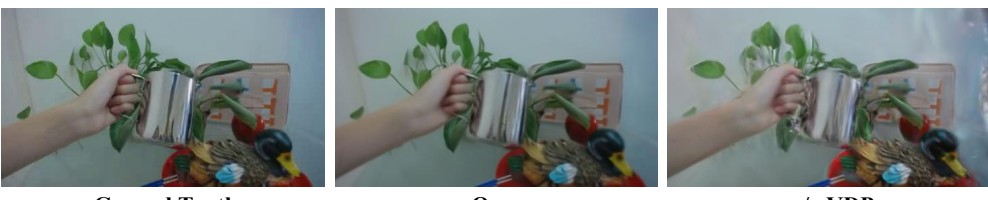

**Ground Truth**            **Ours**            **w/o APP**

Figure 3: Ablation on Appearance Modeling (APP). Comparison of reconstruction results with and without the appearance module. Ours exhibits better alignment and fewer artifacts in static regions.

**Ground Truth**            **Ours**            **w/o VDP**

Figure 4: Removing **VDP** leads to bright blotches near view boundaries due to undertrained Gaussians accumulated at sparsely visible regions.

## 4.2 COMPARED WITH SOTA RESULTS

**Quantitative Comparison.** We compare our approach with Deformable 3DGS (Yang et al., 2023b), 4DGS (Wu et al., 2024), HyperNeRF (Park et al., 2021), NeRF-DS (Yan et al., 2023), GauFre (Liang et al., 2025), MoDec-GS (Kwak et al., 2025). As summarized in Tab. 1, our method consistently achieves the best or second-best performance across most sequences on the NeRF-DS dataset, demonstrating robustness under challenging dynamics and lighting. Averaged over all metrics, it surpasses competing baselines, indicating

| HyperNeRF | | | |
|---|---|---|---|
| Method | PSNR ↑ | SSIM ↑ | LPIPS ↓ |
| **Deformable 3DGS**CVPR 2024 | 24.57 | 0.7641 | 0.2439 |
| **GauFre**WACV 2025 | 23.59 | 0.7486 | 0.2416 |
| **Ours** | 24.61 | 0.7626 | 0.2398 |
| iPhone | | | |
| Method | PSNR ↑ | SSIM ↑ | LPIPS ↓ |
| **Deformable 3DGS**CVPR 2024 | 12.56 | 0.2902 | 0.5896 |
| **GauFre**WACV 2025 | 13.27 | 0.3382 | 0.6206 |
| **Ours** | 13.53 | 0.3391 | 0.6205 |

Table 2: Quantitative results on HyperNeRF (Park et al., 2021) and iPhone (Gao et al., 2022).

superior reconstruction capability. Tab. 2 further reports evaluations on the HyperNeRF and iPhone datasets, where our method remains highly competitive against Deformable 3DGS and GauFre. These results collectively validate the generalizability of our approach across diverse real-world settings involving non-rigid motion and handheld camera trajectories.

**Qualitative Comparison.** We qualitatively evaluate our method on NeRF-DS (Yan et al., 2023), as shown in Fig. 2. This dataset contains dynamic motions and challenging specular effects. Our method achieves high-fidelity results with sharp details, temporally coherent motion, and clean static regions. In particular, it maintains geometric stability and appearance consistency even where dynamic and static elements interact, outperforming existing baselines.

## 4.3 ABLATION STUDY

**Visual Geometry Decomposition Matters.** We conduct an ablation study on the NeRF-DS dataset to quantify the contribution of each component in the **SplitGaussian** framework, as summarized in Tab. 3. Starting from the baseline (a) S1, which separately trains static and dynamic branches, we observe modest reconstruction quality.

Adding S2 in (b) introduces joint optimization between both branches, leading to consistent gains of +0.94 dB in PSNR, +0.0145 in SSIM, and a LPIPS drop of 0.0111, indicating the benefit of mutual spatiotemporal supervision. The introduction of the appearance modeling module in (c) further reduces perceptual distortion (LPIPS ↓ 0.0227), suggesting improved handling of time-varying appearance. In (d), depth-aware pretraining facilitates geometric alignment, contributing an additional +0.42 dB in PSNR and a 0.0163 reduction in LPIPS. Finally, incorporating visibility-driven pruning in (e) yields the best overall performance, achieving a PSNR of 24.03, SSIM of 0.8505, and LPIPS

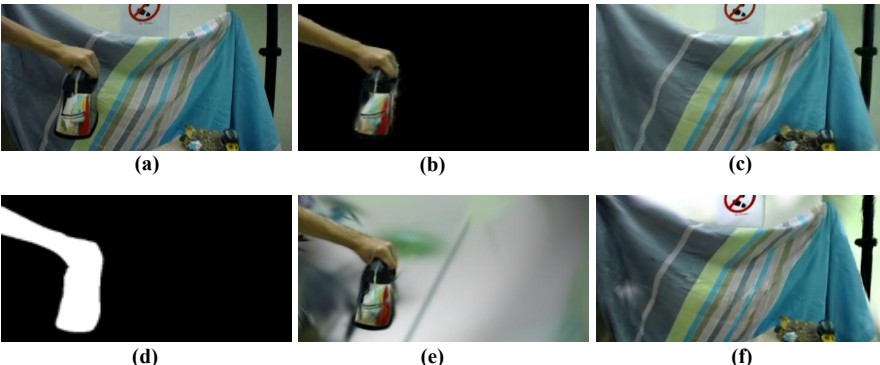

Figure 5: (a) Ground truth. (b) Our dynamic reconstruction. (c) Our static reconstruction. (d) Learned dynamic mask shared by both methods. (e) Dynamic result from GauFRe Liang et al. (2025). (f) Static result from GauFRe Liang et al. (2025).

of 0.1274. These results demonstrate that each module brings measurable improvement, and the full model provides the most stable and perceptually accurate reconstruction.

**Appearance Modeling Improves Static Reconstruction.** Fig. 3 illustrates reconstructions **w/** and **w/o** APP.Removing the appearance module results in noticeable degradation in static regions, especially under subtle lighting changes, leading to artifacts and reduced photorealism. A closer examination reveals that the model with APP can accurately reconstruct fine-grained illumination effects—such as the

| Variant | PSNR ↑ | SSIM ↑ | LPIPS ↓ |
|---|---|---|---|
| (a) S1 (baseline) | 22.41 | 0.8268 | 0.1843 |
| (b) S1+S2 | 23.35 | 0.8413 | 0.1732 |
| (c) S1+S2+APP | 23.36 | 0.8419 | 0.1505 |
| (d) S1+S2+APP+DAP | 23.78 | 0.8457 | 0.1342 |
| (e) (d)+VDP (full) | 24.03 | 0.8505 | 0.1274 |

Table 3: Ablation study on individual components of SplitGaussian on the NeRF-DS dataset.

soft shadow on the static paper surface—while the version without APP fails to capture this detail, producing over-smoothed or inconsistent shading. This highlights the necessity of temporally adaptive appearance modeling for static Gaussians to ensure consistent visual quality across time.

**VDP Improves Peripheral Realism.** Fig. 4 illustrates the impact of removing the Visibility-Driven Pruning (VDP) module. Without VDP, static Gaussians located near the periphery of training views—where visibility is sparse—receive inadequate optimization. As a result, these insufficiently supervised Gaussians accumulate during densification and exhibit artificially elevated opacities. This leads to prominent residual artifacts, particularly along image boundaries, which degrade rendering quality and visual coherence. In contrast, our full method incorporates visibility-aware filtering to suppress low-visibility Gaussians early in training, effectively reducing peripheral noise and producing cleaner, artifact-free reconstructions.

**Beyond Mask-Guided Approaches.** Fig.5 presents a visual comparison between our method and GauFRe Liang et al. (2025). GauFRe effectively integrates occlusion reasoning and mask-based decomposition, enabling basic separation of dynamic and static regions. However, as shown in (e)(f), it occasionally produces blending and structural artifacts under complex motion. In contrast, our approach (b)(c), guided by a learned dynamic mask (d), achieves clearer decomposition in both regions, suggesting enhanced robustness in dynamic-static separation.

## 5 LIMITATIONS AND CONCLUSIONS

While SplitGaussian achieves state-of-the-art performance in dynamic scene reconstruction, its two-stage training pipeline—separately optimizing static and dynamic components before joint appearance modeling—introduces additional computational overhead compared to end-to-end methods. Nevertheless, the proposed decomposition of geometry and appearance, combined with unified spatiotemporal encoding, visibility-driven pruning, and depth-aware regularization, enables temporally coherent and photorealistic reconstruction. Extensive experiments confirm its robustness and generalization across diverse dynamic scenes.

## ETHICS STATEMENT

This work does not involve human subjects, animal experiments, or sensitive personal data. The datasets used (e.g., NeRF-DS, HyperNeRF, iPhone dataset) are publicly available benchmark datasets widely used in computer vision and graphics research and do not contain personally identifiable information. Our method focuses on reconstructing dynamic 3D scenes from monocular video through explicit decomposition of static and dynamic components, and it does not introduce harmful applications. We have carefully reviewed the ICLR Code of Ethics and confirm that this submission complies with its principles regarding fairness, privacy, and research integrity. No potential conflicts of interest exist among the authors.

## REPRODUCIBILITY STATEMENT

To ensure reproducibility, we provide the following resources:(1) All implementation details, including network architectures, hyperparameters, training protocols, and optimization schedules, are described in Section 3 and Section 4, as well as in the Appendix.(2) Theoretical formulations of our SplitGaussian decomposition, including derivations of the static and dynamic components, loss functions, and pruning strategies, are fully detailed in Section 3 and the Appendix.(3) Although fixed random seeds were not enforced, all experiments were repeated multiple times and results are reported as averaged performance to ensure reliability.(4) We use publicly available datasets (NeRF-DS, HyperNeRF, iPhone), which allows for direct benchmarking and verification by other researchers. (5) Our anonymous project page and code are available at https://anonymous.4open.science/w/Anonymous-page-8980/.

## LLM USAGE STATEMENT

Large Language Models (LLMs) were used in this work solely as a general-purpose writing assistance tool—for example, to improve grammar, polish wording, or check technical terminology in the manuscript. LLMs did not contribute to the conception of the research idea, theoretical analysis, experimental design, or interpretation of results. All scientific content, including algorithms, equations, experimental results, and claims, was developed and verified by the authors. No LLM was used to generate novel technical content or to draft substantial portions of the paper. As required by ICLR policy, we confirm that LLMs are not listed as authors, and we take full responsibility for all content under our names.

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

APPENDIX

## A ADDITIONAL QUALITATIVE RESULTS ON HYPERNERF

We further evaluate our method on HyperNeRF (Park et al., 2021), as shown in Figure 6. This dataset covers more complex real-world scenarios, including dynamic motions, challenging lighting conditions, and diverse camera trajectories. Our method achieves perceptually plausible reconstructions with stable geometry and temporally coherent motion, demonstrating its robustness and generalization beyond NeRF-DS.

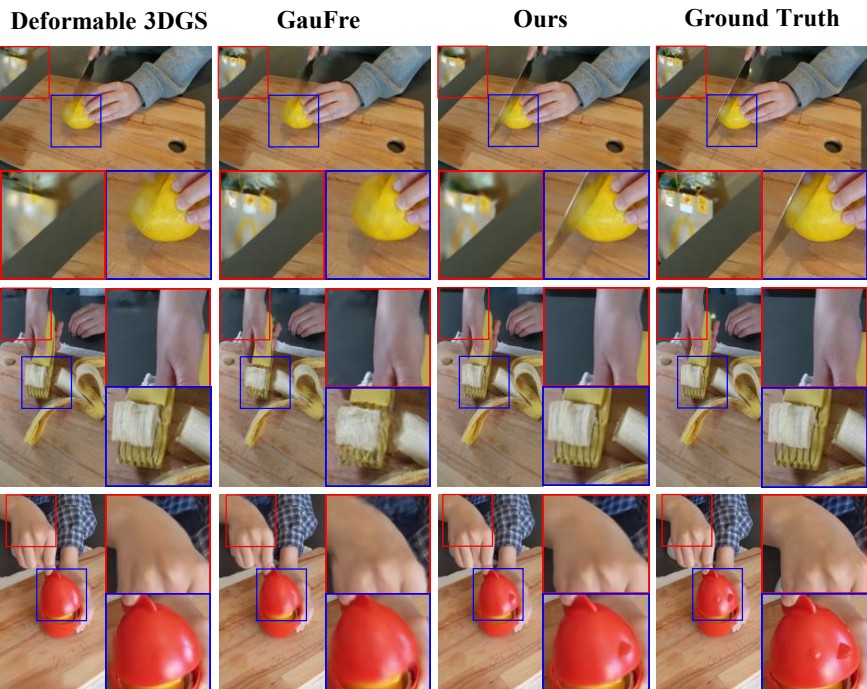

Figure 6: Qualitative results comparison on HyperNeRF (Yan et al., 2023) monocular video dataset.