# OpenReview forum: "SplitGaussian: Reconstructing Dynamic Scenes via Visual Geometry Decomposition"
_ICLR.cc/2026/Conference — ICLR 2026 Conference Withdrawn Submission_

### Official Review · Reviewer_bhUc · 2025-10-20

**Soundness:** 2
**Presentation:** 2
**Contribution:** 1
**Rating:** 2
**Confidence:** 5

**Summary:**

SplitGaussian is a framework for **reconstructing dynamic 3D scenes from monocular videos**, addressing the challenge of jointly inferring **motion, geometry, and appearance** from limited inputs. Unlike previous Gaussian Splatting–based methods that mix static and dynamic elements in a single representation—causing **motion leakage**, **geometric distortion**, and **temporal flickering**—SplitGaussian **explicitly separates** scene representations into **static** and **dynamic** components.

By allowing only the dynamic branch to deform over time while keeping the background geometry fixed, the method **avoids motion artifacts in static regions** and supports **view- and time-dependent appearance refinement**. This **disentangled design** improves **temporal consistency**, **reconstruction fidelity**, and **training stability**, while also **accelerating convergence**. Extensive experiments show that SplitGaussian achieves **state-of-the-art performance** in **rendering quality**, **geometric stability**, and **motion separation**.

**Strengths:**

[S1] Clearly written equations

The equations are presented in a concise and intuitive manner, making it easy for readers to follow the mathematical formulation.
They effectively connect the theoretical design with implementation details.
This clarity significantly enhances the overall readability and technical understanding of the paper.

[S2] Clear and informative figures

The figures visually convey the proposed method and its workflow with high clarity.
They effectively complement the textual explanations and highlight key design components.
This strong visual presentation helps readers quickly grasp the core ideas of the approach.

[S3] Well-organized ablation studies

The ablation experiments are systematically designed to isolate the contribution of each component.
Results are clearly presented and discussed, providing strong empirical justification for the proposed choices.
This structured evaluation enhances the credibility and completeness of the paper’s experimental section.

**Weaknesses:**

[W1] Large overlap with RoDyGS [1].

The overall concept of SplitGaussian shows substantial overlap with RoDyGS, which also separates static and dynamic components of SfM points. Moreover, on the iPhone benchmark—a common evaluation dataset between SplitGaussian and RoDyGS—RoDyGS significantly outperforms SplitGaussian, despite being trained under a pose-free setup. The authors should clarify the key distinctions and contributions of SplitGaussian compared to RoDyGS, which was first published in 2024.

[W2] Lack of technical novelty.

Most of the components presented in this work have already been introduced in prior studies. RoDyGS [1] proposed static–dynamic Gaussian decomposition, while 4DGS [2] introduced a deformation network for Gaussian motion prediction. Furthermore, the concept of APP appears to be a submodule of the time-evolved and view-dependent color modeling already developed in 4DGS [2]. Moreover, using Track Anything to obtain motion mask is already introduced in RoDyGS.

[W3] Missing recent baselines.

For both the iPhone and HyperNeRF benchmarks, the comparisons are limited to a narrow set of baselines. The authors should include additional recent methods such as [1], [2], [3], [4], [5], and [6] to provide a more comprehensive evaluation.

[W4] Lack of qualitative results on iPhone.

No qualitative results are provided for the iPhone benchmark, which is a critical omission. Given that their reported PSNR (13.53) is considerably low, the authors should provide visual evidence and discussion to explain their low performance.

[W5] Missing video demonstrations.

Static image comparisons are insufficient to demonstrate the temporal effectiveness of the proposed approach. The authors should include video results to better highlight the dynamic reconstruction and motion consistency achieved by SplitGaussian.

[W6] Duplicate entries in the bibliography.
The following duplicated entries appear in the references and should be corrected:
- Jonathon Luiten, Georgios Kopanas, Bastian Leibe, and Deva Ramanan. Dynamic 3D Gaussians: Tracking by Persistent Dynamic View Synthesis. In 3DV, 2024a.
- Jonathon Luiten, Georgios Kopanas, Bastian Leibe, and Deva Ramanan. Dynamic 3D Gaussians: Tracking by Persistent Dynamic View Synthesis. In 3DV, 2024b.

[1] RoDyGS: Robust Dynamic Gaussian Splatting for Casual Videos, Jeong et.al., arXiv2024

[2] Real-time Photorealistic Dynamic Scene Representation and Rendering with 4D Gaussian Splatting, Yang et.al., ICLR24

[3] 4D Gaussian Splatting for Real-Time Dynamic Scene Rendering, Wu et.al., CVPR24

[4] MoSca: Dynamic Gaussian Fusion from Casual Videos via 4D Motion Scaffolds, CVPR2025

[5] Shape-of-Motion: 4D Reconstruction from a Single Video, Wang et.al., ICCV2025

[6] RoDynRF: Robust Dynamic Radiance Fields, Liu et.al., CVPR2023

**Questions:**

No questions about the paper.
My major concern is to clarify the difference with previous work and to put more qualitative results including videos.
Although the authors claimed they achieved state-of-the-art perormance, they have omitted recent baselines for comparison.
I hope the authors to add more recent baselines suggested in the weakness section.

---

### Official Review · Reviewer_nwW2 · 2025-10-29

**Soundness:** 3
**Presentation:** 3
**Contribution:** 1
**Rating:** 4
**Confidence:** 5

**Summary:**

The paper first identifies that existing dynamic 3D Gaussian Splatting methods often mix static and dynamic regions, causing motion leakage and geometric distortion. To address this, it proposes a framework called SplitGaussian, which explicitly separates static and dynamic components in the Gaussian representation to decouple motion modeling from background geometry. The method aims to improve temporal consistency and reconstruction stability while maintaining efficient optimization.

**Strengths:**

* The paper identifies a well-defined weakness in prior dynamic 3DGS works (coupled geometry–appearance modeling) and provides a clear, theoretically sound decomposition strategy.

* The paper reports extensive quantitative and qualitative results across multiple datasets, including ablation studies that isolate each module's contribution.

* The paper is easy to follow, with well-organized sections and clear figures that effectively convey the pipeline and improvements.

**Weaknesses:**

While I agree that prior approaches such as Deformable 3DGS, which apply a unified deformation network to both static and dynamic regions, indeed suffer from some problems, my main concern with this paper lies in its insufficient novelty and contribution.

* The proposed solution to the coupling issue primarily relies on using external masks to separate static and dynamic regions and then applying conventional deformation modeling to the dynamic part. This idea has already been explored in many prior works like [1,2]. So I think the design is too simple and does not demonstrate sufficient originality or contribution.

* The *Spatiotemporal Encoding* (Sec 3.2) is quite common in existing dynamic 3DGS implementations, and the setup of the *static and dynamic components* (Sec 3.2) largely follows the standard design used in prior methods. The *VDP* (Sec 3.3) and *DAP* (Sec 3.4) both focus on static regions, while the dynamic part shows little innovation, which does not align well with the paper's claimed core contribution. And the DAP essentially resembles a simple depth loss, which has already been widely used in many prior works and is typically not considered a methodological innovation. It's fine to use existing modules in a reasonable way to solve problems more effectively, but this paper spends too much space describing widely used designs, which weakens its originality.

* For methods with a similar core idea, such as BARD GS, COGS, DeGauss, DynaSplat, and GauFre (mentioned in Remark I), the experiments only compare with GauFre. The paper should also include results against these methods to better show the advantages of the proposed method over similar works.

* The paper should clearly specify the full composition of the final loss function used in training.

In addition, I have several questions regarding the experimental section:

* For the NeRF-DS dataset, what resolution was used during training? Some metrics reported in Table 1 are lower than those provided in the original papers. For example, the PSNR of Deformable 3DGS is 24.11 originally but 23.64 here, and GauFre reports 23.9 originally but 23.54 here. What factors contribute to these discrepancies?

* For the NeRF-DS dataset, MotionGS[3] reports a PSNR of 24.54 and has been open-sourced. Since MotionGS outperforms this work in quantitative results, I would like to know why it was not included as a comparison. What are the key advantages or contributions of this work compared with MotionGS?

* For the HyperNeRF and iPhone datasets, were all scenes used or only a subset? I suggest providing the detailed per scene results in the appendix if there is not enough space in the main paper.

* Could you provide visual comparison results on the iPhone dataset?

[1] MoSca: Dynamic Gaussian Fusion from Casual Videos via 4D Motion Scaffolds. CVPR 2025.

[2] Efficient Gaussian Splatting for Monocular Dynamic Scene Rendering via Sparse Time-Variant Attribute Modeling. AAAI 2025.

[3] MotionGS: Exploring Explicit Motion Guidance for Deformable 3D Gaussian Splatting. NeurIPS 2024.

**Questions:**

See weaknesses.

---

### Official Review · Reviewer_959i · 2025-10-31

**Soundness:** 2
**Presentation:** 3
**Contribution:** 2
**Rating:** 2
**Confidence:** 4

**Summary:**

The proposed SplitGaussian framework introduces a two-branch design that explicitly separates static and dynamic Gaussians while integrating spatiotemporal encoding, deformation networks, and visibility-driven pruning for interpretable and stable reconstructions. Furthermore, the inclusion of depth-aware pretraining and visibility-based pruning enhances geometric stability, accelerates convergence, and improves static reconstruction reliability, effectively overcoming the limitations of prior single-motion-field-based 4DGS models.

**Strengths:**

1. The paper proposes a reasonable two-branch framework that explicitly separates static and dynamic Gaussians. By integrating spatiotemporal encoding, deformation networks, and visibility-driven pruning, the approach provides a clear structure for handling motion and contributes to more stable reconstruction results, even if the overall architectural novelty is moderate.
2. The inclusion of depth-aware pretraining and visibility-based pruning improves geometric stability, accelerates convergence, and enhances static reconstruction reliability. This training design helps mitigate some limitations observed in previous single-motion-field-based 4DGS models, leading to more consistent optimization behavior.

**Weaknesses:**

1. The proposed method heavily relies on the accuracy of the binary mask used to separate static and dynamic regions. Since this mask plays a crucial role in guiding the disentanglement process, the overall reconstruction quality may be highly sensitive to mask quality. Therefore, it is important to demonstrate the robustness of the method across different mask generation models or noise levels. An ablation or sensitivity analysis on mask reliability would strengthen the claims.

2. Lack of comparison with existing static–dynamic decomposition methods. Several prior works (e.g., [1], [2], [3]) also adopt explicit or implicit strategies to separate static and dynamic components during optimization. To fully validate the effectiveness of the proposed disentanglement framework, it is necessary to include both quantitative and qualitative comparisons against these methods, especially in terms of how well each approach preserves static geometry and isolates dynamic motion.

3. The quantitative results reported in the original MoDec-GS [4] paper are higher than those of the proposed method presented in Table 2 of this paper, raising questions about the claimed superiority of the approach. This suggests that the proposed model may not have achieved sufficient generalization capability for monocular 4D scene reconstruction. Additional quantitative comparisons with a broader range of existing methods, particularly on real-world datasets such as iPhone and HyperNeRF, are necessary to more clearly demonstrate the robustness and generalization capability of the proposed framework.

<Minor weakness>
Although this paper addresses the separation between static and dynamic components, it lacks sufficient discussion of existing static–dynamic decomposition methods. In particular, among monocular 4DGS models, prior approaches that separate static and dynamic regions beyond using a single motion field (e.g., [1], [2], [3]) should be clearly described. Including these related works and elaborating on how they motivate the proposed method would help strengthen the overall motivation and positioning of the paper.

**Questions:**

Please refer to the weaknesses mentioned above.

---

### Note · Authors · 2025-11-14

I have read and agree with the venue's withdrawal policy on behalf of myself and my co-authors.